# Shallow distance-dependent triplet energy migration mediated by endothermic charge-transfer

Runchen Lai [1,2], Yangyi Liu[3], Xiao Luo [1], Lan Chen[4], Yaoyao Han[1,5], Meng Lv[3], Guijie Liang[6], Jinquan Chen [3], Chunfeng Zhang [4], Dawei Di [2], Gregory D. Scholes [7], Felix N. Castellano [8] & Kaifeng Wu [1✉]

Conventional wisdom posits that spin-triplet energy transfer (TET) is only operative over short distances because Dexter-type electronic coupling for TET rapidly decreases with increasing donor acceptor separation. While coherent mechanisms such as super-exchange can enhance the magnitude of electronic coupling, they are equally attenuated with distance. Here, we report endothermic charge-transfer-mediated TET as an alternative mechanism featuring shallow distance-dependence and experimentally demonstrated it using a linked nanocrystal-polyacene donor acceptor pair. Donor-acceptor electronic coupling is quantitatively controlled through wavefunction leakage out of the core/shell semiconductor nanocrystals, while the charge/energy transfer driving force is conserved. Attenuation of the TET rate as a function of shell thickness clearly follows the trend of hole probability density on nanocrystal surfaces rather than the product of electron and hole densities, consistent with endothermic hole-transfer-mediated TET. The shallow distance-dependence afforded by this mechanism enables efficient TET across distances well beyond the nominal range of Dexter or super-exchange paradigms.

[1] State Key Laboratory of Molecular Reaction Dynamics and Dynamics Research Center for Energy and Environmental Materials, Dalian Institute of Chemical Physics, Chinese Academy of Sciences, Dalian, Liaoning, China. [2] State Key Laboratory of Modern Optical Instrumentation, College of Optical Science and Engineering, International Research Center for Advanced Photonics, Zhejiang University, Hangzhou, Zhejiang, China. [3] State Key Laboratory of Precision Spectroscopy, East China Normal University, Shanghai, China. [4] National Laboratory of Solid State Microstructures, School of Physics and Collaborative Innovation Center of Advanced Microstructures, Nanjing University, Nanjing, Jiangsu, China. [5] University of the Chinese Academy of Sciences, Beijing, China. [6] Hubei Key Laboratory of Low Dimensional Optoelectronic Materials and Devices, Hubei University of Art and Science, Xiangyang, Hubei, China. [7] Frick Chemistry Laboratory, Princeton University, Princeton, NJ, USA. [8] Department of Chemistry, North Carolina State University, Raleigh, NC, USA. ✉email: kwu@dicp.ac.cn

Molecular spin-triplet states are involved in numerous important applications[1], including but not limited to organic synthesis[2–4], photodynamic therapy[5], light-emitting devices[6,7], and photon upconversion[8–10]. Spin-polarized triplets are also valuable for dynamic nuclear polarization for enhanced nuclear magnetic resonance sensitivity[11] as well as for photo-generating qubits relevant to quantum information science[12]. As dictated by quantum mechanics, however, direct optical excitation of molecules from spin-singlet ground states to triplet excited states is inefficient. Generation of triplets thus relies on triplet energy transfer (TET) from sensitizers, which are molecules or inorganic nanocrystals featuring strong spin-orbital coupling facilitating intersystem crossing[10,13–16].

Besides being practically useful, TET represents a fascinating and fundamental topic in physical chemistry for many decades[17–22]. In Dexter's original formula, the coupling matrix element for TET results from a two-electron exchange integral[17]. Later works have identified additional, important contributions from through-configuration (or so-called super-exchange) interactions[18,23]. This is the formal derivation of the intuitive virtual double-electron transfer that promotes the exchange of triplet photo-excitation—a mechanism that greatly outweighs the Dexter exchange integral in magnitude, but is equally and strongly attenuated as a function of donor–acceptor separation.

More recently it has been established for molecular donor–acceptor systems[24–26], and for nanocrystal-molecule constructs[27–31], that net TET can be achieved by a sequence of essentially uncorrelated exothermic charge-transfer (CT) steps. This mechanism prevails because the coupling matrix element of one-electron transfer is usually larger than that of two concerted electron transfers. The issue with exothermic CT-mediated TET, however, is that it often results in a large energy loss in the sensitization process[27,28].

An alternative mechanism for triplet migration, which has been completely overlooked to date, is stepwise TET mediated by endothermic CT states (Fig. 1). The short-lived, undetectable

nature of endothermic CT states makes it difficult to differentiate this mechanism being operative with respect to Dexter or super-exchange, as they collectively behave like "one-step" TET in terms of measurable spectroscopic features. Nevertheless, concerted and endothermic CT-mediated TET mechanisms should display distinct donor–acceptor wavefunction electronic couplings, manifested in their respective distance-dependences of transfer rates.

In this work, we investigate TET from colloidal CdSe quantum dots (QDs), featuring systematically varied ZnS shell thicknesses, to surface-anchored anthracene molecules. Time-resolved spectroscopy measurements show no evidence for anthracene cation and/or anion formation, excluding exothermic CT-mediated triplet migration. The TET rate decreases with increasing ZnS shell thickness, with rate attenuation clearly following the trend of hole probability density on QD surfaces rather than the product of electron and hole probability densities. This observation contradicts concerted Dexter or super-exchange mechanisms and strongly evidences an endothermic hole-transfer-mediated mechanism. The temperature dependence of the transfer rate further confirms the endothermic hole-transfer process. The shallow distance-dependence of endothermic CT-mediated TET enables efficient triplet migration over donor–acceptor separation beyond Dexter or super-exchange paradigms.

## Results

**Wavefunction criterion for TET mechanisms.** The wavefunction criterion used herein to differentiate TET mechanisms is enlightened by prior studies on long-range intramolecular TET[19,20]. Following Dexter's original formula[17], the electronic coupling matrix element for TET is dominated by a two-electron exchange integral ($K$). Harcourt et al. have later reformulated this matrix element and identified the importance of additional super-exchange terms ($U$)[18,32]. Nonetheless, because both $K$ and $U$ scale with $\langle \Psi_{LU}^{D} | \Psi_{LU}^{A} \rangle \cdot \langle \Psi_{HO}^{A} | \Psi_{HO}^{D} \rangle$, where D and A stand for donor and acceptor and LU and HO stand for lowest unoccupied and

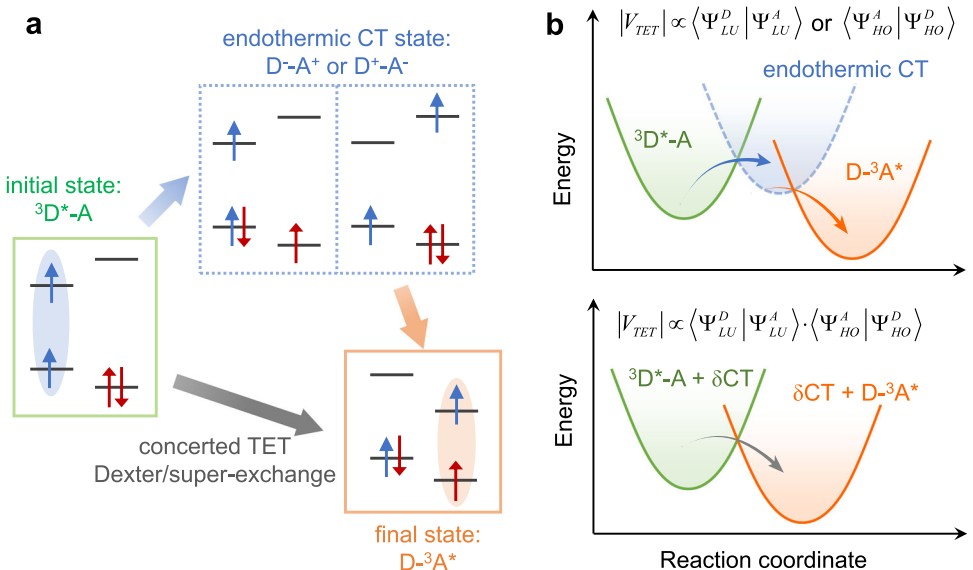

**Fig. 1 Concerted versus endothermic CT-mediated TET. a** Schematic depiction of the initial and final states and intermediate endothermic CT states. $^3D^*$-A and D-$^3A^*$ are the initial and final states, respectively, whereas $D^-$-$A^+$ and $D^+$-$A^-$ are the possible endothermic charge-transfer (CT) states. The red and blue arrows indicate the spins of the electrons in the donor and acceptor, respectively. **b** Concerted (lower) versus endothermic CT-mediated (upper) TET mechanisms drawn in reaction coordinate diagrams. Note that, although not considered in Dexter's original formula, the initial and final states in concerted TET could be mixed with virtual CT states (super-exchange). The electronic coupling matrix elements ($|V_{TET}|$) for concerted and endothermic CT-mediated TET are expressed in terms of donor and acceptor wavefunctions, where D and A stand for donor and acceptor and LU and HO stand for lowest unoccupied and highest occupied molecular orbitals, respectively.

highest occupied molecular orbitals (LUMO and HOMO), respectively, the total matrix element is: $|V_{TET}| \approx -K + U \propto \langle \Psi_{LU}^D | \Psi_{LU}^A \rangle \cdot \langle \Psi_{HO}^A | \Psi_{HO}^D \rangle$. For QD-molecule donor–acceptor systems, it has been established that $\langle \Psi_{LU}^D | \Psi_{LU}^A \rangle$ and $\langle \Psi_{HO}^A | \Psi_{HO}^D \rangle$ are proportional to the amplitudes of electron and hole wavefunctions on the QD surface ($|\Psi_e^S|$ and $|\Psi_h^S|$), respectively[33]. Thus, the scaling relationship for the rate of TET ($k_{TET}$) is: $k_{TET} \propto |V_{TET}|^2 \propto \langle \Psi_{LU}^D | \Psi_{LU}^A \rangle^2 \langle \Psi_{HO}^A | \Psi_{HO}^D \rangle^2 \propto |\Psi_e^S|^2 |\Psi_h^S|^2$; see Supplementary Note 1 for these derivations.

On the other hand, if TET is mediated by an endothermic CT state, the rate of $k_{TET}$ should display distinct dependences on the carrier wavefunctions (Fig. 1b). In this case, the rate of CT from QDs to acceptors ($k_{CT}$) will be proportional to $|\Psi_e^S|^2$ and $|\Psi_h^S|^2$ for electron and hole transfer, respectively. Because the endothermic CT state decays much more rapidly through the second CT step into the product state with respect to its formation (Fig. 1b), the overall rate of TET should be controlled by the first CT step. Therefore, $k_{TET}$ should approximately scale with $|\Psi_e^S|^2$ and $|\Psi_h^S|^2$ in the cases of endothermic electron- and hole-transfer-mediated TET, respectively (Supplementary Note 2).

**QD wavefunction control**. In order to investigate how $k_{TET}$ scales with QD wavefunctions, one needs to tune these wavefunctions in a controllable manner and, ideally, at a constant driving force for TET or CT. This can be achieved using type-I core/shell QDs where the shell has a much larger bandgap than the core and hence acts as a tunneling layer for TET or CT from the photoexcited core[33]. We synthesized CdSe/ZnS core/shell QDs with varying shell thicknesses; see Methods. The zinc blende CdSe core has a diameter of ~2.4 nm and the ZnS shell thickness varies from 0.14 to 1.05 nm, corresponding to 0.5 to 3.9 monolayers of zinc blende ZnS shells (Supplementary Fig. 1). These samples are labeled as CdSe/$x$ZnS ($x$ is the number of ZnS monolayers, 0.27 nm for each monolayer) and their absorption and photoluminescence (PL) spectra are displayed in Fig. 2a. The lowest energy absorption and PL peaks are red-shifted by only ~10 nm (47 meV) from the CdSe core to the CdSe/3.9ZnS, consistent with a type-I band alignment between CdSe and ZnS that strongly localizes the electron and hole in the CdSe core. On the other hand, the PL quantum yield is considerably improved from <10% for the core to 58–94% for the core/shell samples, indicating that the carrier traps on the core surface are effectively passivated by the shell. Because of the low PL quantum yield of the CdSe core-only QDs, TET from them involves not only band-edge excitons but also poorly-understood surface-trapped excitons[34]. For this reason, the core sample will not be evaluated here.

We used a single-band effective mass approximation (EMA) model to quantify the electron and hole envelope wavefunction distributions ($\Psi_e$ and $\Psi_h$) in the core/shell QDs; see Supplementary Note 3 and Table 1 for details. Previous studies have shown that, while being relatively simplified, this model can capture the essence of wavefunction distributions of band-edge states in core/shell QDs[33,35]. The calculated $\Psi_e$ and $\Psi_h$ for CdSe/1.2ZnS are schematically depicted in Fig. 2b. That $\Psi_e$ and $\Psi_h$ can penetrate through the ZnS shell and spread out to the shell surface is a direct manifestation of quantum mechanical tunneling. Figure 2c plots $|\Psi_e^S|^2$ and $|\Psi_h^S|^2$, which are defined as the sum of the respective wavefunction squared on the QD surface divided by the QD surface area, along with their product as a function of the ZnS shell thickness. The variation of these quantities with shell thickness can be quantified using exponential decays $\exp(-\beta d)$, where $\beta$ is the attenuation factor and $d$ the shell thickness. The $\beta$ values are 5.5 and 6.6 nm$^{-1}$, respectively, for the electron and

hole. The $\beta$ value for $|\Psi_e^S|^2 |\Psi_h^S|^2$ is the sum of those of the electron and hole, which is 12.1 nm$^{-1}$. The large difference in their $\beta$ values forms the basis for differentiating concerted and endothermic CT-mediated TET mechanisms using this wavefunction criterion.

**QD-anthracene complexes**. The molecular triplet acceptor used here is a 9-anthracene carboxylic acid (ACA) with a triplet energy of ~1.83 eV[1]. The redox potentials of CdSe/ZnS QDs and ACA molecules were determined by cyclic voltammetry (CV); see Methods and Supplementary Fig. 2. On the basis of these redox potentials, we calculated the energies for electron- and hole-transfer states (QD$^+$-ACA$^-$ and QD$^-$-ACA$^+$, respectively); see Supplementary Note 4. In this calculation, we have included all the relevant Coulomb energy terms, the importance of which has been well explained in our previous study[28]. The calculation indicates that transformation from QD*-ACA to QD$^+$-ACA$^-$ and QD$^-$-ACA$^+$ is energetically uphill by ca. 1.56 and 0.03 eV, respectively. A prior study indeed showed that TET from CdSe QDs to ACA did not involve any detectable charge separation features[16]. We also note that the uncertainties associated with CV measurements and energy calculations could reach hundreds of meV, but the absence of detectable CT states will be directly confirmed using time-resolved spectroscopy.

We assembled QD-ACA complexes using a simple agitation procedure and dispersed them in hexane for spectroscopic studies; see "Methods" section. Prior extensive studies have established that the carboxyl group can bind onto the metal sites on QD surfaces either by replacing original ligands or by filling in unoccupied sites[36–38]. Despite that ACA has negligible solubility in the nonpolar solvent-hexane, the characteristic absorption features of ACA (~320–400 nm) are clearly observed on the absorption spectra of the QD-ACA assemblies (Fig. 2a), suggesting that ACA molecules were successfully anchored onto QD surfaces. By using the absorption spectra and the extinction coefficients of QDs[39] and ACA, we can estimate the average number of ACA molecules per QD ($n_{ACA}$)[16,40]. This number increases from ~21 to ~52 as the shell thickness increased from 0.5 to 3.9 monolayers and scales approximately linearly with the QD surface area (Supplementary Fig. 3). Interestingly, the scaling behavior is similar to the one reported for binding of carboxyl-functionalized pyrene ligands onto CdSe QDs[40]. Further characterizations using a combination of Fourier-transform infrared spectroscopy and gas chromatography–mass spectroscopy suggest that the ACA ligands preferentially bind to unoccupied metal sites on the QD surfaces; see Supplementary Fig. 4 for details.

The static PL spectra of the QD-ACA materials (excited at 460 nm) in comparison to free QDs display different extents of quenching of the QD emission by ACA (Fig. 2a). The quenching efficiency decreased from 83% to 19% as the shell thickness increased from 0.5 to 3.9 monolayers, despite the fact that the number of quenchers ($n_{ACA}$) increased with increasing shell thickness. We exclude the possibility that the PL quenching was induced by introducing new trap states through ligand exchange because QDs surface-functionalized with 1-naphthalene carboxylic acid, a molecule of the same polycyclic aromatic hydrocarbon family but with higher triplet energy (~2.6 eV)[1] than the QD excited state, did not display any noticeable PL quenching (Supplementary Fig. 5). Therefore, the PL quenching behavior in Fig. 2a directly reflects attenuation of the TET rate as a function of shell thickness.

**Time-resolved spectroscopy**. We applied transient absorption (TA) and time-resolved PL (TR-PL) spectroscopy, both having

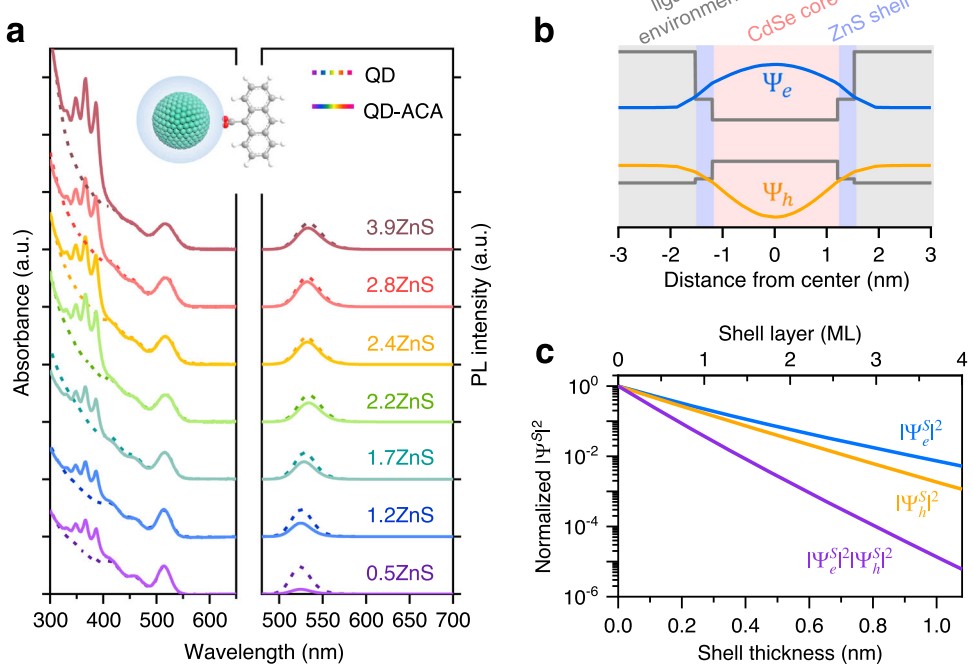

**Fig. 2 Wavefunction control in core/shell QDs. a** Absorption (left) and PL (right) spectra of CdSe/ZnS quantum dots (QDs) with varying shell thicknesses (dashed lines). Similar plots for QD-ACA (9-anthracene carboxylic acid) complexes are in solid lines. The samples are labeled as CdSe/$x$ZnS with $x$ representing the number of ZnS monolayers. **b** Electron (blue) and hole (orange) wavefunction distributions in the lowest excited state of CdSe/1.2ZnS QDs calculated from a single-band effective mass approximation (EMA) model in the single-particle representation. The band alignments are shown by the gray solid lines. **c** Squared electron (blue) and hole (orange) wavefunctions and their product (purple) on CdSe/ZnS QD surface as a function of shell thickness.

sub-ps time resolution, to directly monitor the excited state dynamics in CdSe/ZnS QDs and the QD-ACA constructs; see "Methods" section. In all measurements, the pump pulses selectively excited the QDs and the pump intensities were controlled to ensure the average exciton number per QD was ≤0.05. Results for the free QDs are summarized in Supplementary Fig. 6; these measurements reveal that hole trapping was particularly severe (>90% in amplitude) in core-only QDs and was effectively suppressed by the ZnS shell, which rationalizes the enhancement of PL quantum yield by the shell.

The TA difference spectra of the CdSe/1.2ZnS QD-ACA sample are plotted in Fig. 3a. Accelerated decay of the photoinduced bleach and absorption features of the QDs is accompanied by the gradual formation of an absorption feature centered at 430 nm that can be assigned to the $T_1 \rightarrow T_n$ transitions of ACA triplets ($^3$ACA*)[16], directly evidencing TET from the QDs to surface-anchored ACA. In line with a previous study on CdSe QD-ACA complexes[16], we did not detect any TA features over the range of 600–850 nm where the absorption of anthracene radical cations or anions would be observed (Supplementary Fig. 7). Therefore, the triplet sensitization observed here is not mediated by any long-lived CT states. In Fig. 3b, we compare the TR-PL and TA kinetics, the latter monitored at the XB and the $^3$ACA* feature. The growth of the $^3$ACA* feature clearly tracks the decay of both the TR-PL and XB after 100 ps when carrier trapping becomes negligible. Simultaneous fitting of the TR-PL and TA features reveals an average TET time constant of 10.8 ns; see Supplementary Note 5 and Table 2.

The results above demonstrate that both TA and TR-PL can be used to quantitatively track TET time constants in QD-molecule systems. Figure 3c compares the TR-PL kinetics of CdSe/ZnS QD-ACA samples of varying shell thickness. Clearly, the PL

intensity decay rate attenuates with increasing shell thickness. By fitting all the kinetics in Fig. 3c and comparing their time constants to those of the corresponding free QDs, we obtain TET time constants as well as TET quantum yields for all the samples investigated; see Supplementary Note 5 and Table 2. These TET quantum yields are quantitatively consistent with the PL quenching efficiencies obtained from static PL spectra (Fig. 3d).

**Endothermic CT-mediated TET**. The TET rate constant of each sample determined above was further scaled by the number of acceptors per QD ($n_{ACA}$) in each sample to obtain the intrinsic TET rate constant ($k_{TET}$), i.e., the rate of TET from one QD to one ACA molecule (Supplementary Table 2). Figure 4a presents the plot of normalized $k_{TET}$ as a function of the shell thickness. The attenuation behavior can be best fitted with a $\beta$ value of 6.2 ± 0.2 nm$^{-1}$, which agrees best with that of $\left|\Psi_h^S\right|^2$ (6.6 nm$^{-1}$) and deviates significantly from that of $\left|\Psi_e^S\right|^2\left|\Psi_h^S\right|^2$ (12.1 nm$^{-1}$). This comparison clearly contradicts a concerted two-electron TET model and points to an endothermic hole-transfer-mediated TET mechanism. We note that the $\beta$ value obtained here is similar to that reported in a prior study of charge recombination between the hole in CdSe/ZnS core/shell QDs and a surface-anchored molecular anion radical ($\beta$ = 7.7 ± 0.5 nm$^{-1}$ by excluding the core-only sample)[33] and it is also consistent with that reported for CdSe/CdS QDs ($\beta$ = 4.8 ± 0.6 nm$^{-1}$)[41] considering the differences in hole effective masses and tunneling barrier heights. These consistencies further support that the transfer rates measured here are dominated by a hole-transfer-like process.

We notice that, in addition to the inorganic shells, phenylene bridges are also frequently used to control the distance between QD donors and molecular acceptors[42–45]. The $\beta$ values obtained in those studies, however, are not quantitatively comparable with

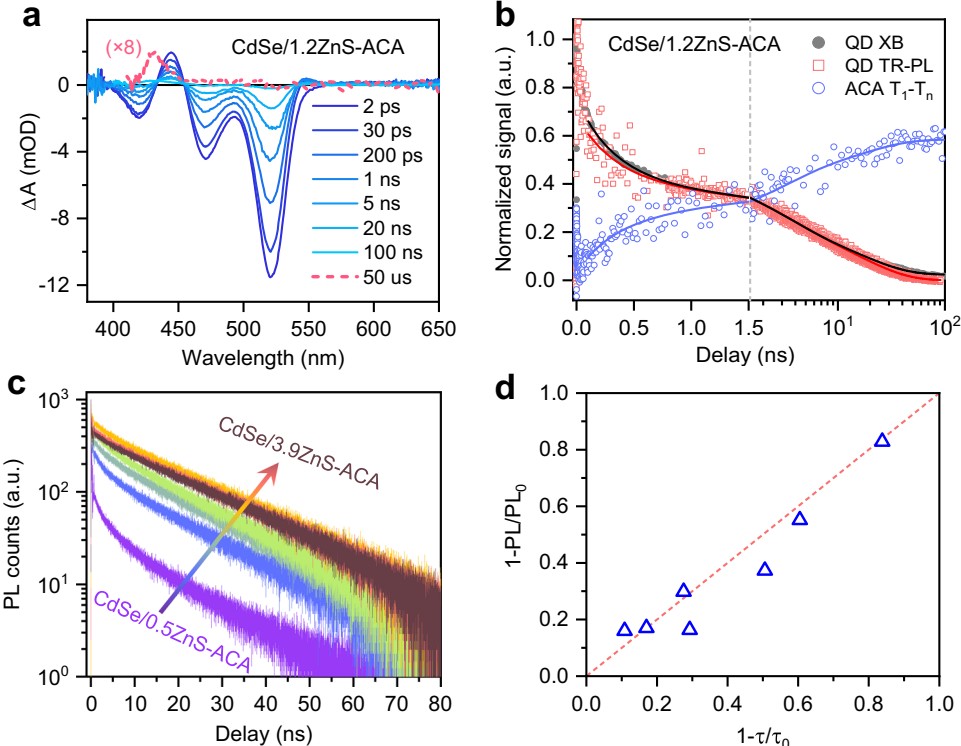

**Fig. 3 TET in QD-ACA complexes. a** Transient absorption (TA) spectra of CdSe/1.2ZnS QD-ACA complexes at indicated delays following 525 nm excitation. Spectrum at a delay of 50 μs is amplified to show the absorption of sensitized ACA triplets. **b** Comparison of exciton bleach (XB; gray solid circles) and time-resolved photoluminescence (TR-PL; red open squares) kinetics of QDs and triplet absorption (open blue circles) of ACA molecules in CdSe/1.2ZnS QD-ACA complexes. The solid lines are multi-exponential fits to the data. **c** TR-PL kinetics of CdSe/ZnS QD-ACA complexes with increasing shell thicknesses (blue to red). **d** Correlation between steady-state (ordinate) and TR-PL (abscissa) quenching efficiencies in QD-ACA complexes. The former and the latter are calculated from $1 - PL/PL_O$ and $1 - \tau/\tau_O$, respectively, where $PL_O$ ($\tau_O$) and PL ($\tau$) are the PL intensities (average lifetimes) of free QDs and QD-ACA complexes, respectively. The red dashed line is a linear scaling relationship.

each other. For example, an early study on CdSe QD-phenylene bridge-anthracene system reported a $\beta$ of 4.3 nm$^{-1}$, but a recent study on a very similar system revealed a $\beta$ of 7.3 nm$^{-1}$. More surprisingly, related work on PbS QD-phenylene bridge-tetracene system reported a $\beta$ of 3.2 nm$^{-1}$ [44], which is even smaller than the $\beta$ values for CdSe QDs despite that the tunneling barrier for the PbS QDs with a lower bandgap should be much higher than that for CdSe QDs. A possible reason for these discrepancies is that the binding geometry of phenylene-functionalized molecules is not well-defined; even if the bridge itself is rigid, the distance between the QD donor and the molecular acceptor can vary with the tilting angle of the molecule with respect to the QD surface normal. Another issue with the phenylene bridge is that the gap of the bridge changes with its length so that the tunneling barrier is not constant. For example, the lowest triplet state energies of benzene and bisphenylene are 3.67 and 2.85 eV, respectively[42]. As such, the $\beta$ value is more of a phenomenological result for these systems. From the above two standpoints, the inorganic ZnS shell we used here is a better choice for a well-defined study of distance-dependent TET from QDs to molecules.

The different TET mechanisms can be further clarified if we compare their rates by considering not only the electronic coupling term but also their respective driving forces. According to non-adiabatic charge-transfer theory, the rate of TET or CT is proportional to the product of the electronic coupling squared ($|V|^2$) and the Franck–Condon-weighted density of states (FCWD) and, at the high-temperature limit, can be expressed as[46,47]:

$$k = \frac{2\pi}{\hbar}|V|^2 \frac{1}{\sqrt{4\pi\lambda k_B T}}\exp\left[-\frac{(\lambda+\Delta G)^2}{4\lambda k_B T}\right].$$ As established above, the distance-dependences are controlled by wavefunctions on QD

surfaces by the factor $|V|^2$. Key parameters in the FCWD are the reorganization energy ($\lambda$) and the TET or CT driving force ($-\Delta G$). The latter has been calculated above. The reorganization energy involves both the inner-sphere ($\lambda_i$) and outer-sphere ($\lambda_o$) components. $\lambda_i$ is mainly contributed by molecules as the electron–phonon coupling in QDs is much weaker compared to the vibronic coupling in small molecules. For the concerted TET process, $\lambda_i$ of ACA was taken as 0.22 eV (half the value for TET between two anthracene molecules)[48] and $\lambda_o$ was set at zero due to unperturbed charge neutrality. For the CT process, $\lambda_i$ and $\lambda_o$ are difficult to determine, but a previous study showed that an extensive set of data on CT from QDs to molecules of similar size as anthracene in nonpolar solvent could be best fitted with a total $\lambda$ of ~0.4 eV[49]. Therefore, the total $\lambda$ for TET and CT processes in the current systems were adopted as 0.22 eV and 0.4 eV, respectively.

With the parameters above, we fit the experimental shell thickness-dependent $k_{TET}$ to different models (Fig. 4b). We find that concerted TET (weakly coupled) and electron transfer (too endothermic) pathways are kinetically incapable to compete with hole transfer. As such, the current experimental system adopts a hole-transfer-mediated TET pathway, where the initial endothermic hole-transfer step is rate-limiting.

In order to further support the endothermic hole-transfer mechanism, we measured the temperature dependence of the transfer rate; see the "Methods" section for details. We performed the measurements for both free CdSe/2.2ZnS QDs and their QD-ACA complexes dispersed in hexane using TR-PL. The temperature-dependent TR-PL traces are presented in Supplementary Fig. 8, from which the TET rates can be obtained using

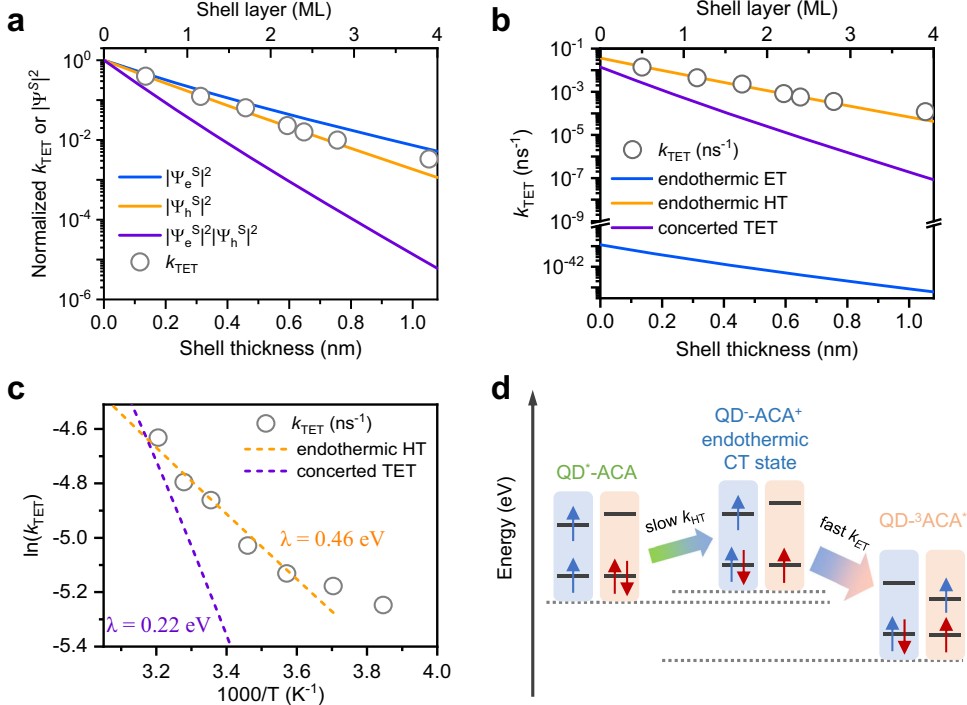

**Fig. 4 Endothermic CT-mediated TET. a** Comparison of experimental shell thickness-dependent TET rates (open circles) and calculated carrier probability density on QD surface (blue, orange, and purple lines for electron, hole, and their products, respectively). These quantities are rescaled by setting the values at 1 for zero shell thickness. **b** Comparison of experimental shell thickness-dependent TET rates (open circles) and calculated TET rates based on concerted (purple line) and endothermic hole-transfer (orange line) and electron transfer (blue line) mediated TET models. **c** Temperature-dependent TET rates in CdSe/2.2ZnS QD-ACA in hexane solution in 260–320 K (open circles) and Marcus theory fitting using an endothermic hole-transfer-mediated TET model with reorganization energy of 0.46 eV (orange dash line). Concerted TET model with reorganization energy of 0.22 eV is also plotted for comparison (purple dash line). **d** Physical picture of endothermic hole-transfer (QD$^-$-ACA$^+$) mediated TET in CdSe/ZnS QD-ACA constructs. $k_{HT}$ and $k_{ET}$ stand for the rates of hole- and electron transfer steps, respectively. Because $k_{HT} \ll k_{ET}$, the overall TET rate is dictated by $k_{HT}$.

the method mentioned above. Figure 4c shows the temperature-dependent TET rates using an Arrhenius-like plot. We fit the data using the Marcus equation discussed above. Note that in the fit we accounted for the slight change of $\Delta G$ with temperature because the QDs showed a slight blue-shift in absorption spectra upon decreasing temperature (Supplementary Fig. 9). The fit reveals a reorganization energy $\lambda$ of 0.46 eV, which is very similar to the one we estimated for hole transfer from our QDs to ACA in hexane (~0.4 eV). In contrast, if using the $\Delta G$ and $\lambda$ (~0.22 eV) for concerted TET, we obtain a temperature-dependent TET rate curve that is much steeper than the experimental results (Fig. 4c). Therefore, the temperature dependence also supports an endothermic hole-transfer-mediated TET model over a concerted one.

Because the hole transfer is an endothermic process, the backward hole-transfer process is faster than forward transfer. In order to guarantee efficient triplet migration, the second electron transfer step following hole transfer should be even faster than backward hole transfer. By assuming reorganization energy of 0.4 eV (the same as hole transfer) and using a driving force of 0.03 eV, the rate of backward hole transfer is ~3.3-fold faster than that of hole transfer. On the other hand, the process of subsequent electron transfer to produce the ACA triplet has a driving force of 0.6 eV (Supplementary Note 4); note that this process is much more energetically favored than direct electron transfer from photoexcited QDs to ground-state ACA, which is a combined result of strong electron-hole Coulomb binding and exchange interaction in ACA molecules, as elaborated in Supplementary Note 4. Also assuming reorganization energy of 0.4 eV, the rate of the triplet-forming electron transfer process is ~70 and ~250-fold

faster than that of hole transfer for the thinnest- and thickest-shell samples, respectively. Thus, the above-mentioned condition for efficient triplet migration can indeed be satisfied. Under this condition, the overall triplet migration process is ultimately controlled by the first hole-transfer step.

The relatively slow formation and fast decay of the endothermic hole-transfer CT state dictate that the population of ACA radical cations cannot be effectively accumulated; see Supplementary Fig. 10 and Note 6 for details. The lack of population accumulation, in combination with the much smaller extinction coefficients of ACA radical cations compared to QDs, provides the rationale why they cannot be observed using transient spectroscopy. Nonetheless, the distance and temperature dependences of the transfer rates are fully consistent with an endothermic hole-transfer-mediated TET model.

A physical picture of the kinetic processes in our CdSe QD-ACA systems is summarized in Fig. 4d. In this scheme, the excitonic state of the QD is drawn as a spin-triplet; this is because the spin of the hole in CdSe-based QDs can be rapidly flipped on a sub-ps timescale[50,51], thus statistically enriching the spin-triplet-like population, although it should be noted that spin is not really a good quantum number in these QDs featuring strong spin-orbit coupling. Starting from this triplet configuration, the nascent hole-transfer CT states (QD$^-$-ACA$^+$) should also be of a triplet character. Flip of the electron spin can result in singlet CT states, but the dominant population should still be triplet CT states. Moreover, in the subsequent electron transfer process, recombination of the singlet CT states to ground-state QD-ACA has a driving force as large as 2.44 eV, which should fall deeply into the Marcus inverted region; in contrast, recombination of the

triplet CT states to QD-$^3$ACA* with a driving force of 0.6 eV is much more kinetically favored. Overall, both spin statistics and kinetic considerations should favor the formation of QD-$^3$ACA* over ground-state QD-ACA.

We note that a few studies proposed a competition between hole transfer and TET from QDs to molecules[52–54], or in another word, recombination of the hole-transfer CT state generates ground-state QD-molecule rather than molecular triplets. However, no direct spectroscopic evidence supporting this competition has been reported. In contrast, more recent studies in such systems have provided a clear correlation between the decay and formation, respectively, of spectroscopic signatures of CT states and molecular triplet states[27,28,30,55–56]. Thus, CT-mediated triplet sensitization is likely a general phenomenon for QD-molecule hybrid materials, as long as the CT states have higher energy than the molecular triplets. Notably, however, all the systems reported to date are based on exothermic CT processes. Our current work is the first one to propose and demonstrate an endothermic CT-mediated triplet sensitization mechanism.

## Discussion

By quantitatively controlling the wavefunction amplitudes on the surfaces of the QD donors, and hence the donor–acceptor electronic couplings, strong evidence has been provided that TET from CdSe/ZnS QD donors to surface-anchored anthracene acceptors is mediated by an endothermic hole-transfer CT state. The current study appears poised to rationalize the collective kinetic data from investigations related to TET from QDs to molecules and, therefore, will serve as a roadmap unifying all future studies in this arena. Moreover, endothermic CT-mediated TET represents a paradigm shift for spin-triplet exciton migration, enabling long-distance triplet sensitization well beyond the capability of Dexter or super-exchange mechanisms. The large band offset between CdSe and ZnS studied here conserves the QD exciton energy during shell coating, thus enabling a well-defined study of the electronic coupling dependence. On the other hand, however, the large valence band offset (~0.85 eV) results in a damping factor ($\beta$) of $6.2 \pm 0.2$ nm$^{-1}$ for the endothermic hole-transfer rate. We expect that, by using core/shell QDs with smaller band offsets (such as CdSe/CdS with conduction and valence band offsets of ~0.3 and 0.45 eV, respectively), we should be able to achieve efficient triplet energy migration across much longer donor–acceptor distances.

## Methods

**Chemicals**. Cadmium oxide (CdO, 99.99% trace metal basis), zinc acetate dihydrate (Zn(Ac)$_2 \cdot$2H$_2$O; 99%), oleic acid (OA; 90% technical grade), selenium powder (Se; 99.999%, 100 mesh), sulfur powder (S; 99.98% trace metals basis), 1-octadecene (ODE, 90% technical grade), 9-anthracene carboxylic acid (ACA, 98 + %), tetra-$n$-butylammonium hexfluorophosphate (98%), and ferrocene (98%) were purchased from Sigma-Aldrich. Anhydrous dichloromethane (99.9%, with molecular sieves) was bought from Macklin Reagents. Hexane, toluene, chloroform, acetone, methanol, ethanol, and ethyl acetate were purchased from Sino-Pharm or Tianjin Damao Reagents. All chemicals were used directly without any further purification.

**Synthesis of CdSe and CdSe/ZnS**. CdSe QDs were synthesized by modifying the hot injection method reported in the literature[57]. Typically, 0.0784 g CdO, 0.8 mL OA, and 10 mL ODE were loaded into a 25-mL three-neck flask, N$_2$ bubbled, and heated to 260 °C to obtain a clear solution. 2 mL of 0.1 M Se-suspension (prepared by dispersing 0.079 g Se powder in 10 mL ODE followed by sonication for 10 min) was injected into the flask to initiate the reaction. After 8 min of reaction at 250 °C, 0.05 mL Se-suspension was added dropwise (15 s per drop), which was followed by another 4 min of reaction. The addition-reaction cycles were conducted until reaching the desired size of CdSe QDs. The reaction was quenched by cooling using an air blower and the cooled mixture was transferred into two centrifuge tubes. 10 mL chloroform, 10 mL acetone, and 5 mL methanol were added to each tube to precipitate the QDs by 3 min of centrifugation at ~5300 × g. The precipitate was dissolved in 5 mL chloroform and further precipitated by 5 mL acetone and 5 mL methanol. The final precipitant was dried and dissolved in hexane for further use.

The purified CdSe QDs were used as seeds to grow CdSe/ZnS core/shell QDs. In a typical reaction, 80 to 300 nmol of seeds were used (calculated using the extinction coefficients reported in ref. [39]). The amounts of zinc and sulfur elements needed in a shelling reaction were calculated based on the seed amount and shell thickness. Zn(Ac)$_2 \cdot$2H$_2$O (four times the calculated amount of zinc elements), 4 equivalent OA and 4 mL ODE were loaded into a 25-mL three-neck flask, N$_2$ bubbled and heated to 250 °C. The mixture was bubbled for 30 min at 250 °C to obtain a clear zinc precursor solution, after which 0.2 mL OA was added into the flask. The mixture was cooled to 200 °C and CdSe seeds in hexane were injected into the flask. After hexane was evaporated, the temperature was set at 250 °C. When the temperature reached 220 °C, 0.2 mL of 0.1 M S-ODE solution (prepared by dissolving 0.032 g S in 10 mL ODE by sonication) was added dropwise into the flask with a speed of 0.02 mL min$^{-1}$, which was followed by 5 min of reaction. The addition-reaction cycles were repeated until a designated amount of sulfur precursor was added. The reaction mixture was cooled to 200 °C, 0.2 mL OA was added, and the mixture was annealed at 200 °C for 2 h before the reaction was quenched by removing the heating mantle and natural cooling to room temperature. The reaction mixture was cooled to 70 °C and transferred to a centrifuge tube. The core/shell QDs were precipitated using 10 mL chloroform, 10 mL acetone, and 5 mL ethanol by 3 min of centrifugation at ~5300 × g. The precipitate was dissolved in 5 mL toluene and further precipitated by 30 mL ethyl acetate. The final precipitant was dried and dissolved in hexane for further use.

**Preparation of QD-ACA complexes**. The QD-ACA complexes were prepared by mixing molecule powder and QD/hexane solution, followed by vigorous stirring for 2 min. The mixture was filtered through a PTFE membrane with 0.22-µm pores to obtain a clear solution of QD-ACA complexes. Since ACA molecules are negligibly soluble in hexane, the molecule absorption features in QD-ACA complex solution arise from the molecules bound to QD surfaces.

**QD characterizations**. Steady-state absorption spectra were taken using an Agilent Cary 60 spectrometer. PL spectra were taken in an Agilent Eclipse fluorescence spectrometer. Photoluminescence quantum yield measurement was carried out using Rhodamine 6G/ethanol solution as a standard (QY = 0.94). Transmission electron microscopy images were taken in a JEOL 2100 F field emission electron microscope operating at 200 kV accelerating voltage.

**Cyclic voltammetry**. CV measurements were conducted on a Model CHI700e electrochemical analyzer with a three-electrode system under an inert gas atmosphere for CdSe/ZnS QDs and ACA molecules. Anhydrous dichloromethane solution containing 0.1 M tetra-$n$-butylammonium hexafluorophosphate was used as the electrolyte. 1 mL of hexane and acetonitrile were used to dissolve QDs and ACA molecules, respectively, before they were added into the electrolyte solution and bubbled in N$_2$. Glassy carbon, Pt-wire, and Ag/AgCl were used as the working, counter, and reference electrodes, respectively. The applied voltage was scanned at a rate of 100 mV s$^{-1}$. The CV curves were calibrated with the ferrocene/ferrocenium (Fc/Fc$^+$) redox couple as an internal standard. The energy level of Fc/Fc$^+$ was taken as −4.8 eV vs vacuum in acetonitrile and −4.86 eV in dichloromethane.

**Fourier-transform infrared (FTIR) spectroscopy**. FTIR measurements were performed on a Thermo Fisher iS50 FTIR spectrometer equipped with iS50 ATR module. Concentrated QD and QD-ACA/hexane solution were drop cast onto the diamond window. The measurements were taken after the solvent evaporated. Spectra of free OA and ACA were collected with the same method, except that the solvent for ACA was acetone.

**Gas chromatography–mass spectroscopy (GC-MS)**. GC–MS measurements were taken with an Agilent 7000D triple quadrupole GC–MS systems. QD and QD-ACA solutions with the same volume and QD concentration were dried under vacuum, then mixed with 4 mL dichloromethane and acetonitrile (5:1 v/v) and 20 µL saturated hydrochloric acid in the glove box. The mixture was vigorously stirred until all QDs were decomposed. DB-WAX and DB-5MS columns were used to run tests for oleic acid and ACA, respectively. Free oleic acid in dichloromethane with concentration from 200 µM to 3 mM and free ACA in acetonitrile from 200 to 600 µM were used as standards to calibrate the instrumental response. All the quantifications used extracted ion chromatogram (EIC) signal integration of $m/z =$ 264.2 (for oleic acid) or 222.1 (for ACA).

**Time-resolved spectroscopy experiments**. Femtosecond and nanosecond TA experiments were based on an Astrella Ti:Sapphire amplifier (Coherent; 800 nm, 70 fs, 6 mJ pulse$^{-1}$ and 1 kHz repetition rate); details were described elsewhere[28]. Briefly, in the femtosecond TA, one part of the 800 nm output was used to pump a TOPAS Optical Parametric Amplifier (OPA) to generate a wavelength-tunable pump beam. Another part with weak intensity was used to generate the white light continuum (WLC) probe beam. The delay between the pump and probe pulses was controlled by a motorized delay stage. The nanosecond TA used the same pump beam, but the WLC was generated by focusing a Nd:YAG laser into a photonic crystal fiber and the pump-probe delay was controlled by a digital delay generator

(CNT-90, Pendulum Instruments). Samples were placed in 1-mm airtight cuvettes and were constantly stirred during measurements. Femtosecond PL upconversion was measured using a home-built fluorescence upconversion system; details were described elsewhere[58]. Briefly, the PL of the sample was collected by a lens and focused into a BBO crystal together with an 800 nm gate beam to generate the up-converted signal via sum-frequency-generation (SFG). The up-converted photons were detected by the spectrometer. Samples were held in a continuously spinning UV quartz disk during measurements.

**Temperature-dependent experiments.** Temperature-dependent absorption spectra and PL decays were collected for the CdSe/2.2ZnS QDs and their QD-ACA complexes dispersed in hexane. The samples were sealed in 1-mm cuvettes and loaded in liquid nitrogen-cooled cryostat (Oxford MicrostateHe). A second temperature sensor was attached to the cuvette in order to accurately record the local temperature of the cuvette. Steady-state absorption spectra were recorded using a fiber-coupled spectrometer (QEpro, Ocean Optics). A broadband tungsten-halogen lamp (Thorlabs) was used as the light source. PL decays were measured by a TCSPC set-up with a time resolution of ~110 ps (PicoHarp 300, Picoquant). Because the PL of CdSe/2.2ZnS QD-ACA showed negligible decay within ~150 ps (as measured by femtosecond PL upconversion at room temperature), the time resolution of TCSPC should be sufficient for the measurements.

## Data availability

The experiment data that support the findings of this study are available from the corresponding author upon reasonable request.

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

## Acknowledgements

K.W. acknowledges financial support from the National Natural Science Foundation of China (21975253), the Strategic Pilot Science and Technology Project of the Chinese Academy of Sciences (XDB17010100), and the Ministry of Science and Technology of China (2018YFA0208703). R.L. acknowledges financial support from China Postdoctoral Science Foundation (2019M651155). F.N.C.-t. acknowledges support from the Air Force Office of Scientific Research (FA9550-18-1-0331). D.D. acknowledges the National Key R&D Program of China (2018YFB2200401). K.W. and R.L. acknowledge useful discussions with Dr. John Philbin and Prof. Eran Ranani regarding the QD wavefunctions.

## Author contributions

K.W. conceived the ideas and designed the project. R.L. prepared the samples. R.L., Y.L., M.L., G.L., and J.C. performed the time-resolved spectroscopy measurements. R.L. and X.L. performed CV measurements. R.L., L.C., D.D., and C.Z. performed temperature-dependent measurements. R.L. and K.W. analyzed the data. K.W., R.L., Y.H. G.D.S., and F.N.C.-t. analyzed the energy transfer mechanisms. K.W. wrote the paper with contributions from all authors.

## Competing interests

The authors declare no competing interests.
