## [Peer Review File · Nature Communications]

REVIEWER COMMENTS

Reviewer #1 (Remarks to the Author):

The manuscript describes the investigation on the mechanism of Dexter triplet-triplet energy transfer from CdSe quantum dots (QDs) to molecular acceptors. The authors mediated the wavefunction leakage of the electron and hole through controlling the thickness of the ZnS shell. The quantitative measurement of the TET rate at different electron and hole densities on the QD surface suggests that TET is correlated to the hole. The authors infer that the TET process is mediated by the endothermic hole transfer process. They point out that this mechanism is consistent with the observation of efficient TET across a long distance. The TET process involving semiconductor nanoparticles is an emerging field because of both fundamental interest and potential applications in energy conversion. Understanding the TET mechanism is essential for enhancing the efficiency of this process. Specifically, long-distance exciton migration through a TET mechanism is highly desirable in energy conversion devices. Although this is interesting research when comparing with previous research in the field, the mechanism described in the manuscript seems to represent a relatively special scenario, and the authors might stretch their theory to an unsuitable system. It will need more experimental results to support the conclusion and careful analysis to clearly define the applicable scope. My comments are summarized in the following four points.

1) An earlier study of TET (Li et al., JPCL, 2016) between CdSe QDs and anthracene has estimated a damping coefficient for through-bond TET, $\beta=0.43 \text{ \AA}^{-1}$, which is much smaller than the number measured by the authors. This finding has already suggested shallow distance-dependent TET. Can the author comment on the difference of TET through ZnS comparing to through phenyl rings?

2) Researchers also studied the efficiency of TET from CdSe QDs to different anthracene derivatives. (Xia, et al., ChemComm, 2017) They employed dithiocarbamate linkers as the anchoring group, which is known to promote hole transfer. However, in the 2017 paper, the hole transfer seems to have a detrimental effect on TET. The different findings here seem to suggest that the CdSe/ZnS system employed by the authors may not reflect the universal nature of TET from CdSe QDs.

3) The authors claim that the TET is mediated by endothermic hole transfer based on the estimation of energy levels. I trust the authors' calculations and measurements. However, it will be more direct to prove the point with a temperature-dependence experiment. If the rate of TET shows a positive correlation with the temperature, the concerted TET mechanism can be ruled out because the latter should be an exothermic process.

4) The author used CdSe/CdS QDs to demonstrate the long-distance TET. The band energy alignment between CdSe and CdS is very different from that of CdSe/ZnS. The energy offset between the conduction bands of CdSe and CdS is small, leading to a lower electron transfer energy barrier compared to that between CdSe and ZnS. Specifically, with thick CdS shell, it is even possible to shift the band alignment from Type I to Type II. In this case, whether the TET is still mediated by hole transfer (as it possibly happened in the CdSe/ZnS system) is unclear. In fact, if the damping coefficient remains the same magnitude as that through ZnS ($\beta = 6.2 \text{ nm}^{-1}$), it will be almost

impossible to carry out efficient TET across a distance of 2.8 nm, given the TET rate measured for 0.5L of ZnS (0.16Å)

Reviewer #2 (Remarks to the Author):

This is an interesting study that attempts to rationalize TET from QD sensitizers to surface bound scene molecules. The studied the distance dependence from the core QD through a wide-bandgap shell to surface bound molecules by varying the shell thickness and using ultrafast spectroscopy to measure the TET rates. The authors find that the rate decreases less fast than one might expect from a Dexter TET mechanism. They propose a two step mechanism that relies on a endothermic charge transfer step (in this case hole transfer) followed by a down hill electron transfer step. The measurements are interesting and have some significance in designing such systems that rely on triplet energy transfer. A few clarifications might help.

The authors state that the CT state is undetectable because it's formation is slow and the subsequent electron transfer to form the triplet state is fast. This makes some sense to me. But can the authors calculate what the number of oxidized species would be in their experiments. It seems that detecting that species should be possible. The hole transfer here is probably fast since it's really not up-hill that much. While the second step might be slower than what's estimated here. Alternatively it would be nice if the authors could show an expected temperature dependence or vary the hole transfer driving force here so that that oxidized species could be detected. the authors could also vary the dielectric environment in an attempt to slow down the electron transfer step. A more direct observation (other than a distance dependence) would be nice.

The ligand coverage and ligand binding mode is rather poorly characterized here. Some of the control experiments do support the authors claims but given the current status of QD-ligand binding modes in the literature I would have though the authors could do a better job. Does the ligand density vary across the samples. How are the number of ligands per QD determined?

Reviewer #3 (Remarks to the Author):

This article presents a combination of spectroscopic, electrochemical and theoretical evidence that TET from a particular anthracene ligand to a well-chosen core/shell CdSe nanocrystal proceeds via an incoherent stepwise process with hole transfer creating a transient Charge Separated State. Overall, I

find the arguments fairly convincing and I believe most of the conclusions. I do have a few concerns that need to be addressed before publication:

1) First and foremost, I don't see that they have really proven that the step is endothermic. They have mainly shown that the intermediate is transient (due to the fact that it doesn't show up in their TA spectra). But that says nothing about it being endothermic. If they want to prove endothermicity, I think there has to be a temperature dependent study here.

2) On a related note, I don't think the theory calculations on the band offsets (showing uphill values of 1.56 and 0.03 eV, respectively) have much value. The theory is heavily based on assumptions of where the energy levels lie, and these numbers just recapitulate those assumptions.

3) In terms of the mechanism, I am a bit concerned about the second step. They get a good fit for HT fitting the experimental data, meaning it is the rate determining step. But strangely, it is *faster* than electron transfer (primarily because ET is computed to be highly endothermic). But then in the second step, ET somehow becomes highly exothermic and the rate increases by 44 orders of magnitude. I understand that some of this comes from the breakup of the e-h binding energy in the first step, followed by regaining some of that energy in the second step. But I don't see how the effect is quite that big. Can the authors explain? In any case, the primary problem here is that even after this huge shift, the ET rate does not appear to be fast enough. According to the TA data, the CS intermediate must live for a very short time (<2 ps) because it is not visible in the TA. But the computed rate of ET is on the tens of ps timescale. So the HT state *should* be visible in the first few TA traces. This needs to be explained.

Reviewer comments in black, our responses in red and revisions in blue

REVIEWER COMMENTS

Reviewer #1 (Remarks to the Author):

The manuscript describes the investigation on the mechanism of Dexter triplet-triplet energy transfer from CdSe quantum dots (QDs) to molecular acceptors. The authors mediated the wavefunction leakage of the electron and hole through controlling the thickness of the ZnS shell. The quantitative measurement of the TET rate at different electron and hole densities on the QD surface suggests that TET is correlated to the hole. The authors infer that the TET process is mediated by the endothermic hole transfer process. They point out that this mechanism is consistent with the observation of efficient TET across a long distance. The TET process involving semiconductor nanoparticles is an emerging field because of both fundamental interest and potential applications in energy conversion. Understanding the TET mechanism is essential for enhancing the efficiency of this process. Specifically, long-distance exciton migration through a TET mechanism is highly desirable in energy conversion devices. Although this is interesting research when comparing with previous research in the field, the mechanism described in the manuscript seems to represent a relatively special scenario, and the authors might stretch their theory to an unsuitable system. It will need more experimental results to support the conclusion and careful analysis to clearly define the applicable scope. My comments are summarized in the following four points.

Response: We thank the reviewer very much for his/her insightful comments on our work. Below we provide our point-to-point responses and revisions to address his/her remaining concerns.

1) An earlier study of TET (Li et al., JPCL, 2016) between CdSe QDs and anthracene has estimated a damping coefficient for through-bond TET, $\beta=0.43 \text{ \AA}^{-1}$, which is much smaller than the number measured by the authors. This finding has already suggested shallow distance-dependent TET. Can the author comment on the difference of TET through ZnS comparing to through phenyl rings?

Response: We thank the reviewer for reminding us this relevant paper. First, we would like to note that the β factor in our work is 6.2 nm^{-1} (i.e., 0.62 \AA^{-1}), and therefore, the β factor of the JPCL paper is just slightly smaller than the one obtained. Also, we notice that the same group has recently reported a β factor of 0.72 \AA^{-1} in their most recent work on essentially the same CdSe QD-phenylene bridge-anthracene system (J. Am. Chem. Soc. 2020, 142, 17581). More surprisingly, their related work on PbS QD-phenylene bridge-tetracene system reported a β factor of 0.32 \AA^{-1} (ACS Appl. Mater. Interfaces 2020, 12, 36558), which is even shallower than the factors for CdSe QDs despite that the tunneling barrier for the PbS QDs with a lower bandgap should be much higher than CdSe QDs. So it seems like that the phenylene bridge can

introduce a quite large uncertainty in the β factor. A possible reason is that the binding geometry of phenylene-functionalized anthracene is not well-defined; even if the bridge itself is rigid, the distance between the QD donor and the molecular acceptor can vary with the tilting angle of the molecule with respect to the QD surface normal. Another issue with the phenylene bridge is that the gap of the bridge actually changes with its length (i.e., the tunneling barrier is not a constant). For example, the lowest triplet state energies of benzene and bisphenylene are 3.67 and 2.85 eV, respectively. Therefore, for these systems, the β factor is more of a phenomenological result. From these two standpoints, the inorganic ZnS shell we used here is a better choice for a well-defined study of distance-dependent TET from QDs to molecules.

The reasons why the system in the JPCL paper also displayed shallow distance dependence ($\beta \sim 0.43 \text{ \AA}^{-1}$) are complicated. But at least one reason is that the tunneling barrier actually becomes shallow when the bridge is bisphenylene (energy offset ~ 0.5 eV). In contrast, the energy offset between our CdSe QD and ZnS shell reaches ~ 1.8 eV. As such, for our system, an endothermic charge transfer mediated model has to be invoked to explain a shallow distance dependence of the TET rates.

Revision: On page 11, we add the following paragraph to discuss the above contents: “We notice that, in addition to the inorganic shells, phenylene bridges are also frequently used to control the distance between QD donors and molecular acceptors.⁴²⁻⁴⁵ The β values obtained in those studies, however, are not quantitatively comparable with each other. For example, an early study on CdSe QD-phenylene bridge-anthracene system reported a β of 4.3 nm^{-1} , but a recent study on a very similar system revealed a β of 7.3 nm^{-1} . More surprisingly, a related work on PbS QD-phenylene bridge-tetracene system reported a β of 3.2 nm^{-1} ,⁴⁴ which is even smaller than the β values for CdSe QDs despite that the tunneling barrier for the PbS QDs with a lower bandgap should be much higher than that for CdSe QDs. A possible reason for these discrepancies is that the binding geometry of phenylene-functionalized molecules is not well-defined; even if the bridge itself is rigid, the distance between the QD donor and the molecular acceptor can vary with the tilting angle of the molecule with respect to the QD surface normal. Another issue with the phenylene bridge is that the gap of the bridge changes with its length, so that the tunneling barrier is not a constant. For example, the lowest triplet state energies of benzene and bisphenylene are 3.67 and 2.85 eV, respectively.⁴² As such, the β value is more of a phenomenological result for these systems. From the above two standpoints, the inorganic ZnS shell we used here is a better choice for a well-defined study of distance-dependent TET from QDs to molecules.”

2) Researchers also studied the efficiency of TET from CdSe QDs to different anthracene derivatives. (Xia, et al., ChemComm, 2017) They employed dithiocarbamate linkers as the anchoring group, which is known to promote hole transfer. However, in the 2017 paper, the hole transfer seems to have a detrimental effect on TET. The different findings here seem to suggest that the CdSe/ZnS system employed by the authors may not reflect the universal nature of TET from CdSe QDs.

Response: We thank the reviewer again for reminding us this relevant paper. Indeed, in the ChemComm paper, Xia et al. observed much lower photon upconversion efficiencies with anthracene dithiocarbamate (ADTC) ligands compared to anthracene carboxylate (ACA) ligands. But it is important to note that the efficiencies of TET from CdSe QDs to ADTC ligands are comparable with ACA ligands (see Fig. 3 in that paper). In that paper, the authors have found that the fluorescence quantum yields of the ADTC ligands are much lower than those of ACA ligands (see Fig. 4a in that paper). Therefore, the authors stated that the low photon upconversion efficiencies associated with ADTC ligands were due to the fast nonradiative quenching of the sensitized ADTC triplet states, which hindered efficient TET from them to the DPA annihilators in the solution.

Admittedly, in that paper, the authors also mentioned another possibility that the dithiocarbamate group should promote hole transfer from QDs to ADTC ligands, which could compete with TET and hence lower the TET efficiencies, but no evidence was provided to support this competition. The statement was made because hole transfer mediated TET had not been established yet for QD-molecule systems at that time. More recent works by our and other groups have clearly demonstrated that charge transfer should mediate rather than compete with TET from QDs to molecules (see, e.g., Nat. Commun. 2020, 11, 28; *J. Am. Chem. Soc.* **2020**, *142*, 11270-11278; *J. Am. Chem. Soc.* 2020, *142*, 4723-4731), as long as the energy of the charge transfer state is higher than that of the molecular triplet.

Revision: On page 14-15, we add the following paragraphs to discuss the relationship between CT and TET:

“A physical picture of the kinetic processes in our CdSe QD-ACA systems is summarized in Fig. 4d. In this scheme, the excitonic state of the QD is drawn as a spin-triplet; this is because the spin of the hole in CdSe-based QDs can be rapidly flipped on a sub-ps timescale,^{50,51} thus statistically enriching the spin-triplet population. Starting from this triplet configuration, the nascent hole transfer CT states (QD⁻ACA⁺) should also be of a triplet character. Flip of the electron spin can result in singlet CT states, but the dominant population should still be triplet CT states. Moreover, in the subsequent electron transfer process, recombination of the singlet CT states to ground-state QD-ACA has a driving force as large as 2.44 eV, which should fall deeply into the Marcus inverted region; in contrast, recombination of the triplet CT states to QD-³ACA* with a driving force of 0.6 eV is much more kinetically favored. Overall, both spin statistics and kinetic considerations should favor the formation of QD-³ACA* over ground-state QD-ACA.

We note that a few studies proposed a competition between hole transfer and TET from QDs to molecules,⁵²⁻⁵⁴ or in another word, recombination of the hole transfer CT state generates ground-state QD-molecule rather than molecular triplets. However, no direct spectroscopic evidence supporting this competition has been reported. In contrast, more recent studies in such systems have provided clear correlation between the decay and formation, respectively, of spectroscopic signatures of CT states and molecular triplet states.^{27,28,30,55,56} Thus, CT-mediated triplet sensitization is likely a

general phenomenon for QD-molecule hybrid materials as long as the CT states have higher energy than the molecular triplets. However, all the systems reported to date are based on exothermic CT processes and our current work is the first one to propose and demonstrate an endothermic CT-mediated triplet sensitization mechanism.”

3) The authors claim that the TET is mediated by endothermic hole transfer based on the estimation of energy levels. I trust the authors' calculations and measurements. However, it will be more direct to prove the point with a temperature-dependence experiment. If the rate of TET shows a positive correlation with the temperature, the concerted TET mechanism can be ruled out because the latter should be an exothermic process.

Response: We thank the reviewer for this very good suggestion. We have now performed a temperature-dependent measurement to further support the endothermic hole transfer mediated TET mechanism. But before we elaborate on our new data, we would like clarify that the both endothermic hole transfer mediated TET and exothermic concerted TET should be activated with increasing temperature. According to Marcus theory, thermal activation brings the reactant and product states to a temporary degenerate state (the crossing point between the reaction and product surfaces in Fig. R1). So regardless of the charge transfer/TET reaction being endothermic or exothermic, the rate should increase with temperature.

Figure R1. Schematic depiction of the activation energy (ΔG^\ddagger) in charge/triplet energy transfer reactions.

Nevertheless, according to Fig. R1, the height of activation barrier indeed changes with the endothermicity/exothermicity of the reaction, which offers an opportunity to differentiate between hole transfer mediated TET and concerted TET in our system. We measured temperature-dependent TET between CdSe/2.2ZnS QDs and surface-anchored ACA ligands using time-resolved PL. Free QDs were also measured under the same conditions. The TET rates were obtained by taking the difference between free QDs and QD-ACA complexes and are plotted in Fig. R2.

Figure R2. Temperature dependent TET rates in CdSe/2.2ZnS QD-ACA in hexane solution in 260-320 K (open circles) and Marcus theory fitting using endothermic hole transfer mediated TET (orange dash line). Concerted TET (purple dash line) model is also plotted for comparison.

We fit the data in Fig. R2 using the Marcus equation. Note that in the fit we accounted for the slight change of ΔG with temperature, because the QDs showed slight blue-shift in absorption spectra upon decreasing temperature. The fit reveals a reorganization energy of 0.46 eV, which is very similar to the one we estimated for hole transfer from our QDs to ACA in hexane (~ 0.4 eV). In contrast, if we use the ΔG and reorganization energy (~ 0.22 eV) for concerted TET, we obtain a temperature-dependent TET rate curve that is much steeper than the experimental results (Fig. R2). Therefore, the temperature dependence data also support an endothermic hole transfer mediated TET model rather than a concerted one.

Revision: On page 13, we add the following paragraph to discuss the temperature dependence:

“In order to further support the endothermic hole transfer mechanism, we measured the temperature dependence of the transfer rate; see Methods for details. We performed the measurements for both free CdSe/2.2ZnS QDs and their QD-ACA complexes dispersed in hexane using TR-PL. The temperature-dependent TR-PL traces are presented in Supplementary Fig. 8, from which the TET rates can be obtained using the method mentioned above. Fig. 4c shows the temperature-dependent TET rates using an Arrhenius-like plot. We fit the data using the Marcus equation discussed above. Note that in the fit we accounted for the slight change of ΔG with temperature, because the QDs showed slight blue-shift in absorption spectra upon decreasing temperature (Supplementary Fig. 9). The fit reveals a reorganization energy λ of 0.46 eV, which is very similar to the one we estimated for hole transfer from our QDs to ACA in hexane (~ 0.4 eV). In contrast, if

using the ΔG and λ (~ 0.22 eV) for concerted TET, we obtain a temperature-dependent TET rate curve that is much steeper than the experimental results (Fig. 4c). Therefore, the temperature dependence also supports an endothermic hole transfer mediated TET model over a concerted one.”

In addition, Fig. R2 is included as a new Fig. 4c in the revised paper and the original temperature-dependent PL decay traces and sample absorption spectra are provided in the new Supplementary Figs. 8 and 9, respectively. Experimental details for the temperature-dependent measurements are added to the Methods section.

4) The author used CdSe/CdS QDs to demonstrate the long-distance TET. The band energy alignment between CdSe and CdS is very different from that of CdSe/ZnS. The energy offset between the conduction bands of CdSe and CdS is small, leading to a lower electron transfer energy barrier compared to that between CdSe and ZnS. Specifically, with thick CdS shell, it is even possible to shift the band alignment from Type I to Type II. In this case, whether the TET is still mediated by hole transfer (as it possibly happened in the CdSe/ZnS system) is unclear. In fact, if the damping coefficient remains the same magnitude as that through ZnS ($\beta = 6.2 \text{ nm}^{-1}$), it will be almost impossible to carry out efficient TET across a distance of 2.8 nm, given the TET rate measured for 0.5L of ZnS (0.16 \AA)

Response: We thank the reviewer for this very insightful comment. Admittedly, We have not provided enough information to support the endothermic hole transfer mediated TET for the CdSe/CdS QDs which has a different band alignment with the CdSe/ZnS QDs. Here we qualitatively explain the reasons accounting for the efficient TET observed in thick-shell CdSe/CdS QDs.

The first one is related to the band alignment between CdSe and CdS, as pointed by the reviewer. With thick CdS shells, the CdSe/CdS QDs form a so-called quasi-type II electronic structure, with the hole confined in the CdSe core and the electron delocalized among the CdSe core and the CdS shell. For this reason, the lifetime of the photogenerated exciton in CdSe/CdS QDs increases with the shell thickness because of a reduced electron-hole overlap. The longer lifetime window is beneficial for TET. The second reason is that the valence band offset between CdSe and CdS (~ 0.45 eV) is also much lower than that between CdSe and ZnS (~ 0.85 eV). As a result, a smaller β factor can be expected for hole tunneling through CdS than ZnS.

Nevertheless, quantitative measurement of the TET mechanisms for CdSe/CdS QDs is probably beyond the scope of this work. For the sake of clarity, we have decided to remove this part of data from the manuscript. The mechanism study for CdSe/ZnS QDs alone is the focus of the current work.

Reviewer #2 (Remarks to the Author):

This is an interesting study that attempts to rationalize TET from QD sensitizers to surface bound scene molecules. The studied the distance dependence from the core

QD through a wide-bandgap shell to surface bound molecules by varying the shell thickness and using ultrafast spectroscopy to measure the TET rates. The authors find that the rate decreases less fast than one might expect from a Dexter TET mechanism. They propose a two step mechanism that relies on an endothermic charge transfer step (in this case hole transfer) followed by a down hill electron transfer step. The measurements are interesting and have some significance in designing such systems that rely on triplet energy transfer. A few clarifications might help.

Response: We thank the reviewer very much for his/her kind comments on our work. Below we provide our point-to-point responses and revisions to clarify the issues he/she raised.

The authors state that the CT state is undetectable because its formation is slow and the subsequent electron transfer to form the triplet state is fast. This makes some sense to me. But can the authors calculate what the number of oxidized species would be in their experiments. It seems that detecting that species should be possible.

Response: We thank the reviewer for this constructive suggestion. Per this suggestion, we simulated the temporal evolution of QD^*-ACA , QD^-ACA^+ (i.e. CT state) and $QD-^3ACA^*$ species using the coupled rate equations of eqs. (R1) -**Error! Reference source not found.**, where k_1 and k_2 are hole and electron transfer rates, respectively, and k_{-1} is the backward hole transfer rate. The value of k_1 was from the experimental data, whereas k_2 and k_{-1} were estimated according to the Marcus equation by assuming that the reorganization energies were the same for these CT processes.

$$\frac{d[QD^* - A]}{dt} = -k_1[QD^* - A] + k_{-1}[CT] - k_r[QD^* - A] \quad (R1)$$

$$\frac{d[CT]}{dt} = k_1[QD^* - A] - (k_{-1} + k_2)[CT] \quad (R2)$$

$$\frac{d[QD - ^3A^*]}{dt} = k_2[CT] \quad (R3)$$

We chose CdSe/0.5ZnS QDs for the simulation, because the ratio between k_1 and k_2 is largest for this sample, i.e. the most possible one for us to detect the ACA cations. The temporal evolution of the transient species is presented in Figure R3. From the simulation, the QD^-ACA^+ CT species contributes at most 1% of the total species population at a time delay of 0.1-0.2 ns. Further considering the orders-of-magnitude difference in the extinction coefficients of QDs and ACA cations, it is technically impossible to detect the ACA cations on TA spectra.

Figure R3. Simulated temporal evolution of relative populations of QD^*-ACA , QD^*-ACA^+ and $QD-^3ACA^*$ in $CdSe/0.5ZnS-ACA$ complexes in the first (a) 100 ns and (b) 2 ns.

Revision: We add Fig. R3 and related discussion as the new Supplementary Fig. 10 and Note 6, respectively. These contents are quoted on Page 14 of the main text as: “The relatively slow formation and fast decay of the endothermic hole-transfer CT state dictates that the population of ACA radical cations cannot be effectively accumulated; see Supplementary Fig. 10 and Note 6 for details.”

Alternatively it would be nice if the authors could show an expected temperature dependence or vary the hole transfer driving force here so that that oxidized species could be detected. the authors could also vary the dielectric environment in an attempt to slow down the electron transfer step. A more direct observation (other than a distance dependence) would be nice.

Response: We thank the reviewer for these further suggestions to confirm the hole-transfer-mediated TET. First, we would like to note that definitive evidence for exothermic hole transfer mediated TET has been provided in our prior study (Nat. Commun. 2020, 11, 28) using CsPbBr₃ QD-tetracene complexes by correlating the decay of the QD hole signal with formation of the tetracene cations as well as the subsequent decay of the QD electron signal with the delayed formation of the tetracene triplets. Therefore, varying the hole transfer driving force and detecting the cation species would be a repetition of our prior study. The focus of the current work is how to evidence an endothermic hole transfer mediated TET when the cation species is not detectable.

As to the suggestion of varying the dielectric environment, this is indeed a good way to tune the energetics of the CT state and hence hole and electron transfer rates. But we don't have many choices for the solvent here. Nonpolar solvents are required for stable dispersion of the QDs and for stable anchoring of the ACA molecules onto QD surfaces. For these reasons, we adopted the suggestion of measuring temperature dependence to further support the endothermic hole transfer mediated TET. Because this suggestion has also been made by Reviewer 1, details can be found in our response to his/her comment 3).

The ligand coverage and ligand binding mode is rather poorly characterized here. Some of the control experiments do support the authors claims but given the current status of QD-ligand binding modes in the literature I would have though the authors could do a better job. Does the ligand density vary across the samples. How are the number of ligands per QD determined?

Response: We thank the reviewer very much for this insightful comment on QD-ligand binding. The ACA ligand is functionalized with a carboxyl group, and the binding modes of carboxyl-functionalized molecules onto QDs have already been extensively characterized in the literature (see, e.g., Owen et al. *J. Am. Chem. Soc.* 2013, 135, 18536; Tang et al. *Nano Lett.* 2014, 14, 3392; Beard et al. *Nat. Commun.* 2017, 8, 15257). It is well known that the carboxyl group binds to the metal sites on QD surfaces by replacing original ligands or by filling in empty sites. For this reason, the binding modes of ACA ligands on QDs were not studied again here using, e.g. NMR. Herein, by using a combination of FTIR and GC-MS (Supplementary Fig. 4), we have confirmed that filling in empty metal sites on QD surfaces is the major way of QD-ACA binding. Also, we would like to note that binding of the same or similar ligands onto CdSe-based QDs for TET studies has already been established by Castellano et al. (*Science* 2016, 351, 369; *Nat. Chem.* 2018, 10, 225) and by Tang et al. (*Nano Lett.* 2015, 15, 5552; *J. Am. Chem. Soc.* **2017**, 139, 9412).

As for the number of ligands per QD, this number was determined based on the absorption spectra of QD-ACA complexes and the extinction coefficients of QDs and ACA. Because the ACA molecules have a negligible solubility in the nonpolar solvent hexane, we can assign the calculated molecule/QD ratio to the average number of molecules bound to each QD. This is also a standard method used in many prior studies on TET/CT between QDs and molecules. According to our calculation, the number of ACA molecules on each QD roughly scales with the surface area of the QD (Supplementary Fig. 3); thus the ligand density remains roughly constant among the samples. Moreover, the scaling coefficient (number of ACA per nm²) determined for samples is ~0.78, which is very similar to the one reported in *Nat. Chem.* 2018, 10, 225. This agreement further suggests that the number of ligands per QD in our samples has been correctly estimated.

Revision: On page 7-8, we have expanded the paragraph on QD-ligand binding:

“We assembled QD-ACA complexes using a simple agitation procedure and dispersed them in hexane for spectroscopic studies; see Methods. Prior extensive studies have established that the carboxyl group can bind onto the metal sites on QD surfaces by either replacing original ligands or by filling in unoccupied sites.³⁶⁻³⁸ Despite that ACA has a negligible solubility in the nonpolar solvent-hexane, the characteristic absorption features of ACA (~320-400 nm) are clearly observed on the absorption spectra of the QD-ACA assemblies (Fig. 2a), suggesting that ACA molecules were successfully anchored onto QD surfaces. By using the absorption spectra and the extinction coefficients of QDs³⁹ and ACA, we can estimate the average number of ACA molecules per QD (n_{ACA}).^{16,40} This number increases from ~21 to ~52 as the

shell thickness increased from 0.5 to 3.9 monolayers and scales approximately linearly with the QD surface area (Supplementary Fig. 3). Interestingly, the scaling behavior is similar to the one reported for binding of carboxyl-functionalized pyrene ligands onto CdSe QDs.⁴⁰ Further characterizations using a combination of Fourier-transform infrared spectroscopy and gas chromatography-mass spectroscopy suggest that the ACA ligands preferentially bind to unoccupied metal sites on the QD surfaces; see Supplementary Fig. 4 for details.”

Reviewer #3 (Remarks to the Author):

This article presents a combination of spectroscopic, electrochemical and theoretical evidence that TET from a particular anthracene ligand to a well-chosen core/shell CdSe nanocrystal proceeds via an incoherent stepwise process with hole transfer creating a transient Charge Separated State. Overall, I find the arguments fairly convincing and I believe most of the conclusions. I do have a few concerns that need to be addressed before publication:

Response: We thank the reviewer very much for his/her kind comments on our work. Below we provide our point-to-point responses and revisions to address his/her remain concerns.

1) First and foremost, I don't see that they have really proven that the step is endothermic. They have mainly shown that the intermediate is transient (due to the fact that it doesn't show up in their TA spectra). But that says nothing about it being endothermic. If they want to prove endothermicity, I think there has to be a temperature dependent study here.

Response: We thank the reviewer for this constructive suggestion. In fact, the same suggestion has also been made by Reviewers 1 and 2. We have performed a temperature dependent study to support the endothermic hole transfer mechanism. Details can be found in our response to Reviewer 1's comment 3).

2) On a related note, I don'y think the theory calculations on the band offsets (showing uphill values of 1.56 and 0.03 eV, respectively) have much value. The theory is heavily based on assumptions of where the energy levels lie, and these numbers just recapitulate those assumptions.

Response: We thank the reviewer for this comment. It is true that the charge transfer driving forces rely heavily on the “single-particle” energy levels determined from CV measurements. But Coulomb energy terms have to be incorporated to correctly calculate the charge transfer driving forces, because these interactions can be quite strong for small-size systems such as QDs and molecules studied here. This point has been well explained in prior related studies (Lian et al. Nano Lett. 2014, 14, 1263; Kamat et al. Proc. Natl. Acad. Sci. 2011, 108, 29; Wu et al. Nat. Commun. 2020, 11, 28).

For example, CV measurements indicate that the oxidation potential energies of

ground state QDs and ACA molecules are -5.94 and -5.82 eV, respectively. Without considering Coulomb energy terms, these numbers would suggest that hole transfer from QDs to ACA is exothermic with a driving force of 0.12 eV. This is obviously inconsistent with the experimental result (undetectable ACA cation species). By including Coulomb energy terms, the reaction becomes endothermic with a driving force of -0.03 eV.

Also, we note that while the reviewer suggested that the calculation did not have much value, we have encountered in our prior paper submissions reviewers who paid particular attention to these Coulomb energy corrections.

Revision: On page 7, we add the following sentence to emphasize these Coulomb terms:

“In this calculation, we have included all the relevant Coulomb energy terms, the importance of which has been well explained in our previous study.²⁸”

3) In terms of the mechanism, I am a bit concerned about the second step. They get a good fit for HT fitting the experimental data, meaning it is the rate determining step. But strangely, it is **faster** than electron transfer (primarily because ET is computed to be highly endothermic). But then in the second step, ET somehow becomes highly exothermic and the rate increases by 44 orders of magnitude. I understand that some of this comes from the breakup of the e-h binding energy in the first step, followed by regaining some of that energy in the second step. But I don't see how the effect is quite that big. Can the authors explain?

Response: We thank the reviewer for this comment. The reviewer has raised a very important question. Clearly, we have two electron transfer reactions here to be clarified. One is ET from QD* to ACA to form ACA⁻ (ET1), and the other is ET from QD⁻ to ACA⁺ to form ³ACA* (ET2). The difference between these two lies not only in the e-h binding energy (~0.8 eV) in the molecule (as notified by the reviewer), but also in the exchange interaction in ACA. Specifically, we need to also understand the difference between ETs from QD⁻ to ACA⁺ to form ³ACA* and to form ¹ACA*. The difference for the driving forces of these two is ~1.27 eV, which is the singlet-triplet splitting in ACA arising from a strong electron exchange interaction. The combination of e-h binding, exchange interaction and other minor terms make the driving forces for ET1 and ET2 differ by ~2.16 eV. These details have been included in our calculations in Supplementary Note 4, particularly eqs. S31 and S32.

In any case, the primary problem here is that even after this huge shift, the ET rate does not appear to be fast enough. According to the TA data, the CS intermediate must live for a very short time (<2 ps) because it is not visible in the TA. But the computed rate of ET is on the tens of ps timescale. So the HT state **should** be visible in the first few TA traces. This needs to be explained.

Response: We thank the reviewer for this comment. We would like to note that the hole transfer state is invisible not because it is shorter-lived than our instrument response, but rather is from a kinetic consideration. Specifically, the rate of hole transfer is much slower than that of electron transfer; as a result, the charge separated

state (QD^- - ACA^+) cannot accumulate sufficient population for spectroscopic detection. This point has been elaborated in our response to Reviewer 2's comment 1 (Fig. R3).

Revision: On page 13-14, we add the following paragraphs to clarify the two questions:

“Because the hole transfer is an endothermic process, backward hole transfer process is faster than forward transfer. In order to guarantee efficient triplet migration, the second electron-transfer step following hole transfer should be even faster than backward hole transfer. By assuming a reorganization energy of 0.4 eV (the same as hole transfer) and using a driving force of 0.03 eV, the rate of backward hole transfer is ~ 3.3 -fold faster than that of hole transfer. On the other hand, the process of subsequent electron transfer to produce the ACA triplet has a driving force of 0.6 eV (Supplementary Note 4); note that this process is much more energetically favored than direct electron transfer from photoexcited QDs to ground-state ACA, which is a combined result of strong electron-hole Coulomb binding and exchange interaction in ACA molecules, as elaborated in Supplementary Note 4. Also assuming a reorganization energy of 0.4 eV, the rate of the triplet-forming electron transfer process is ~ 70 and ~ 250 -fold faster than that of hole transfer for the thinnest- and thickest-shell samples, respectively. Thus, the above-mentioned condition for efficient triplet migration can indeed be satisfied. Under this condition, the overall triplet migration process is ultimately controlled by the first hole-transfer step.

The relatively slow formation and fast decay of the endothermic hole-transfer CT state dictates that the population of ACA radical cations cannot be effectively accumulated; see Supplementary Fig. 10 and Note 6 for details. The lack of population accumulation, in combination with the much smaller extinction coefficients of ACA radical cations compared to QDs, provides the rationale why they cannot be observed using ultrafast transient spectroscopy. Nonetheless, the distance and temperature dependences of the transfer rates are fully consistent with an endothermic hole transfer mediated TET model.”

REVIEWERS' COMMENTS

Reviewer #1 (Remarks to the Author):

The temperature-dependent experiment resolved the doubt about endothermic hole transfer. As a mechanistic investigation, the manuscript satisfies the standard of publishing.

However, the fact that the authors took out the last part from the original manuscript weakened their argument that CT can induce long-distance TET. In the CdSe/ZnS system discussed in the current version, efficient TET can only be achieved with less than 1.5 layers of ZnS (estimated from Fig.2, regarding >50% PL quenching as efficient TET), in which QD and the molecular acceptor are separated by <0.4 nm. This is not a very impressive distance for carrying out TET. In fact, the approach implemented in the current manuscript does not demonstrate advantages over the previous research where oligophenylenes are used as the bridge between the QD and anthracene. (e.g., *J. Phys. Chem. Lett.* 2016, 7, 1955 and *J. Am. Chem. Soc.* 2020, 142, 17581) Similar efficiency of TET through the molecular bridge with a distance close to that is reported by the authors were reported. It is worthwhile to point out that oligophenylene is a rigid molecular bridge and known for enhancing the electronic coupling between the donor and the acceptor to facilitate long-range Dexter processes. The current manuscript can be recognized as a nice following up contribution to the group's previous work on CT mediated TET in the QD-molecule system. (e.g., *Nat. Commun.* 2020, 11, 28; *J. Am. Chem. Soc.* 2020, 142, 11270; and *J. Am. Chem. Soc.* 2020, 142, 4723). Hence, the current manuscript is more suitable for a journal that has a more specific scope in the field of physical chemistry.

Reviewer #2 (Remarks to the Author):

The authors have done a good job responding to all reviewers comments and suggested improvements and I can recommend publication.

Reviewer comments in black, and our responses in blue

Reviewer #1 (Remarks to the Author):

The temperature-dependent experiment resolved the doubt about endothermic hole transfer. As a mechanistic investigation, the manuscript satisfies the standard of publishing.

However, the fact that the authors took out the last part from the original manuscript weakened their argument that CT can induce long-distance TET. In the CdSe/ZnS system discussed in the current version, efficient TET can only be achieved with less than 1.5 layers of ZnS (estimated from Fig.2, regarding >50% PL quenching as efficient TET), in which QD and the molecular acceptor are separated by <0.4 nm. This is not a very impressive distance for carrying out TET. In fact, the approach implemented in the current manuscript does not demonstrate advantages over the previous research where oligophenylenes are used as the bridge between the QD and anthracene. (e.g., J. Phys. Chem. Lett. 2016, 7, 1955 and J. Am. Chem. Soc. 2020, 142, 17581) Similar efficiency of TET through the molecular bridge with a distance close to that is reported by the authors were reported. It is worthwhile to point out that oligophenylene is a rigid molecular bridge and known for enhancing the electronic coupling between the donor and the acceptor to facilitate long-range Dexter processes. The current manuscript can be recognized as a nice following up contribution to the group's previous work on CT mediated TET in the QD-molecule system. (e.g., Nat. Commun. 2020, 11, 28; J. Am. Chem. Soc. 2020, 142, 11270; and J. Am. Chem. Soc. 2020, 142, 4723). Hence, the current manuscript is more suitable for a journal that has a more specific scope in the field of physical chemistry.

Revision: We thank the reviewer for his/her further comments on the distance dependence of TET from core/shell QDs. Because the CdSe/CdS system has not been rigorously studied yet in terms of the TET mechanism, we think it is better to present it in our future work. Here we only add a short outlook at the end of the discussion part:

“The large band offset between CdSe and ZnS studied here conserves the QD exciton energy during shell coating, thus enabling a well-defined study of the electronic coupling dependence. On the other hand, however, the large valance band offset (~0.85 eV) results in a damping factor (β) of $6.2 \pm 0.2 \text{ nm}^{-1}$ for the endothermic hole transfer rate. We expect that, by using core/shell QDs with smaller band offsets (such as CdSe/CdS with conduction and valence band offsets of ~0.3 and 0.45 eV, respectively), we should be able to achieve efficient triplet energy migration across much longer donor-acceptor distances.”

Reviewer #2 (Remarks to the Author):

The authors have done a good job responding to all reviewers comments and suggested improvements and I can recommend publication.

Revision: We thank the reviewer very much for recommending publication of our paper.